# Understanding Gel-Powers: Exploring Rheological Marvels of Acrylamide/Sodium Alginate Double-Network Hydrogels

**DOI:** 10.3390/molecules28124868

**Published:** 2023-06-20

**Authors:** Shi-Chang Wang, Shu-Tong Du, Saud Hashmi, Shu-Ming Cui, Ling Li, Stephan Handschuh-Wang, Xuechang Zhou, Florian J. Stadler

**Affiliations:** 1College of Materials Science and Engineering, Shenzhen Key Laboratory of Polymer Science and Technology, Guangdong Research Center for Interfacial Engineering of Functional Materials, Shenzhen University, Shenzhen 518055, China; 2060341020@email.szu.edu.cn (S.-C.W.); mosmcui@mail.scut.edu.cn (S.-M.C.); liandzero@163.com (L.L.); 2College of Chemistry and Environmental Engineering, Shenzhen University, Shenzhen 518055, China; 2060221013@email.szu.edu.cn (S.-T.D.); stephanwang@scut.edu.cn (S.H.-W.); xczhou@szu.edu.cn (X.Z.); 3Department of Polymer & Petrochemical Engineering, NED University of Engineering & Technology, Karachi 75270, Pakistan; saudhashmi@cloud.neduet.edu.pk; 4The International School of Advanced Materials, School of Emergent Soft Matter, South China University of Technology, Guangzhou 511442, China

**Keywords:** acrylamide, sodium alginate, double network hydrogel, large amplitude oscillatory shear, Fourier transform rheology

## Abstract

This study investigates the rheological properties of dual-network hydrogels based on acrylamide and sodium alginate under large deformations. The concentration of calcium ions affects the nonlinear behavior, and all gel samples exhibit strain hardening, shear thickening, and shear densification. The paper focuses on systematic variation of the alginate concentration—which serves as second network building blocks—and the Ca^2+^-concentration—which shows how strongly they are connected. The precursor solutions show a typical viscoelastic solution behavior depending on alginate content and pH. The gels are highly elastic solids with only relatively small viscoelastic components, i.e., their creep and creep recovery behavior are indicative of the solid state after only a very short time while the linear viscoelastic phase angles are very small. The onset of the nonlinear regime decreases significantly when closing the second network (alginate) upon adding Ca^2+^, while at the same time the nonlinearity parameters (Q_0_, I_3_/I_1_, S, T, e_3_/e_1_, and v_3_/v_1_) increase significantly. Further, the tensile properties are significantly improved by closing the alginate network by Ca^2+^ at intermediate concentrations.

## 1. Introduction

Hydrogels are polymeric materials that can swell and retain large amounts of water in their three-dimensional cross-linked network structure, resulting in solid-like properties in the steady state [1]. Synthetic hydrogels have better mechanical properties than naturally occurring ones and can be chemical or physical gels depending on their cross-linking method [2]. Chemical gels are formed through polymerization or cross-linking of monomers, while physical gels are formed through intermolecular interaction forces, while in most cases, entanglements do not play an important role, as the entanglement molar masses at the concentrations chosen are usually significantly higher than the molar mass between the crosslinks. This is easily visible from the fact that in most cases the precursor solutions of the polymers (gels being made up of polymer chains and not polymerized directly from a monomer) are more or less Newtonian with viscosities like water or oil, i.e., between 1 mPas and 1 Pas at room temperature [3,4,5,6,7,8,9,10,11,12,13,14,15,16,17,18,19,20,21,22,23,24]. However, chemical gels are typically fragile and are permanently damaged when subjected to high strain, resulting in poor mechanical strength [25,26].

In 2003, double-network (DN) hydrogels were introduced by Gong et al. [27] using a two-step sequential radical polymerization method. The unique entangled network structure of DN hydrogels presents the potential for improved mechanical properties. DN hydrogels consist of two networks, one rigidly cross-linked to form a rigid network and one ductile and loosely cross-linked, leading to a nonlinear effect between the networks that provides high strength [27]. The irregularities in the double network structure contribute to better mechanical properties [28,29].

There has been a recent surge in interest in DN hydrogels with controllable shape deformation and diverse network structures. Various modifications have been proposed to enhance their mechanical properties, electrical properties, and fatigue fracture resistance, resulting in wide applications in a wide array of fields, such as environmental and electrochemical engineering, biomedical devices, sensors, and tissue engineering [30,31,32,33].

Double network (DN) hydrogels and interpenetrating polymer network (IPN) hydrogels are two distinct types of hydrogels with different structural characteristics. DN hydrogels consist of two interlocked networks, resulting in a highly durable and tough material, while IPN hydrogels are formed by at least two polymer networks that are interpenetrated but not necessarily interlocked, leading to enhanced mechanical properties. Gong et al. [27] clarified that DN hydrogels are composed of two networks with different structures and properties, while IPN hydrogels are partially interleaved at the molecular scale, with at least one network chemically cross-linked [34,35]. DN hydrogels were later classified as a special type of IPN hydrogel. Despite their similarities, DN and IPN hydrogels exhibit variations in mechanical strength and behavior due to their distinct structural characteristics. However, IPN hydrogels have yet to improve significantly mechanical strength compared to their original networks due to their soft and rubbery consistency [36]. An overview discussing the classification of hydrogels based on a physical or chemical cross-linking preparation route together with the physical properties was published in 2017 [37].

Mechanical properties are crucial in the application of hydrogels and cannot be overlooked [38]. The comprehension of the rheological properties of hydrogels is vital in controlling their performance. Hydrogel research focuses on the correlation between network structure and hydrogel properties and the relationship between high strain and fracture [39]. The science of understanding the flow and deformation of materials is the study of rheology. Rheology can define the microstructure of soft materials [40], analyze changes in dynamic mechanical properties, and describe structural features. In polymer processing, rheology bridges the molecular structure and processing properties. The guidance of rheology is widely used in industrial polymer chemistry and processing technology [41].

Rheological characterization includes steady-state shear and dynamic mechanical tests (DMT) that measure viscosity, yield stress, zero shear viscosity, and thixotropy at varying shear rates. Dynamic oscillatory shear testing is commonly used to study complex fluids and soft materials, such as polymer melts and solutions, block copolymers, biopolymers, polyelectrolytes, surfactants, suspensions, and emulsions. DMT uses the storage (G′) and loss (G″) moduli to represent elastic and viscous behavior, respectively, under shear. High G′ relative to G″ indicates solid-like behavior, and low G′ relative to G″ indicates liquid-like behavior. The loss factor parameter (tan δ) is an indicator of the gel threshold [42,43,44,45]. DMT includes small amplitude oscillatory shear (SAOS) for linear viscoelastic response and large amplitude oscillatory shear (LAOS) for nonlinear material response observed at larger strain amplitudes (>0.1–30% depending on the sample) [7,43,46,47,48,49,50]. 

Linear viscoelasticity has limitations in the understanding of the rheological properties of complex fluids. In processing operations, large and fast deformations result in non-linear behavior. Non-linear viscoelasticity is essential for distinguishing complex fluids with similar micro- or nano-structures. Matching non-linear parameters improves consistency between molecular theory, constitutive equations, and experiments [51,52,53,54,55]. Steady-state viscosity measurements are useful but limited, particularly for high deformation rates. LAOS testing is a valuable alternative for a range of complex fluids and soft materials, allowing independent variation of strain amplitude and frequency and easy generation and control [46,56,57,58,59,60].

LAOS and SAOS require appropriate selection of strain amplitude (γ_0_) and frequency (ω) for experimental input, but LAOS output analysis differs from that of SAOS due to material response becoming nonlinear under sufficiently large strain amplitudes [46,53,56,57,58,59,60,61,62]. LAOS has been explored over the past two decades in biopolymer gel rheology, allowing for deeper insights into microstructural changes under large deformation [63]. LAOS data analysis is crucial for interpreting complex rheological behavior, and various methods, such as Lissajous curves and Fourier transforms, have been used. LAOS characterization of biopolymers provides a better understanding of rheological behavior and can simulate actual polymer processing conditions [40,41,47,55,60,61,64,65,66,67,68].

The polyacrylamide (PAAM)/sodium alginate (SA) composite hydrogel is a typical DN structure [69], consisting of two interpenetrating polymer networks: a covalently cross-linked PAAM network and an alginate network formed by ionic interactions [69]. Linear polysaccharides extracted from brown algae cell walls include β-d-mannuronic acid (M) and α-l-guluronic acid (G) and are alginate salts. In aqueous solution, G/M-blocks of alginate chains cross-link with bivalent cations (e.g., Ca^2+^) to form an alginate hydrogel, (+), which can be explained by the formation mechanism of the “egg-box” structure [70]. Amongst other properties, the swelling and mechanical properties of this hydrogel are pH-dependent [71]. PAAM is typically obtained by free-radical polymerization of acrylamide (AM) by the irreversible formation of covalent bonds [72]. To obtain the DN hydrogel, the neutral monomer AM is first cross-linked to form the first network. Then, the polyelectrolyte SA is introduced to prepare the second network with rigid and strong properties, which can penetrate the first network [34]. Therefore, DN hydrogels have better mechanical properties than gels with a single network [73,74].

The PAAM network in the DN structure is elastic, while the alginate network is brittle and fractures by the breaking of ionic bonds [69]. The fracture process in the alginate network dissipates energy through hysteresis [75]. Gong′s group investigated PAAM single network hydrogels using mechanical and rheological characterization and proposed constitutive equations for their tensile and frictional behavior [76,77,78,79,80,81,82,83].

Wu et al. [84] investigated the structure and frictional behavior of PAAM/SA ion-covalent hybrid and sequential double network hydrogels on glass in water, NaCl, and CaCl_2_. They utilized a rotational rheometer to measure the friction between the samples and the glass substrate. The results indicated that the frictional stress of the PAAM/SA gel could be managed by the adsorptive elastic hydrodynamic friction of the covalently cross-linked PAAM network and the repulsive hydrodynamic friction of the minor SA ion network. They proposed a qualitative model to describe the effect of the ionic environment solution on the hybrid gel friction behavior. The findings demonstrated that the ionic environment solution significantly impacted the ion cross-linking density, swelling degree, and charge shielding of the SA ion network at the gel-glass interface.

Stadler’s group [53,57] investigated the linear viscoelasticity, nonlinear rheology, and microstructure of composite hydrogels made of sodium alginate (SA) and poly (N-isopropylacrylamide-co-4-vinylphenylboronic acid) (NIBA) by varying the SA and NIBA content. They studied the interaction between NIBA and SA and the viscoelasticity of SA/NIBA blends as a function of NIBA concentration, using LAOS to obtain structural information. The mechanism of gelation and microstructure of SA/NIBA gels were elucidated. They found that SA and NIBA blends were in a sol state under alkaline and neutral conditions and formed hydrogen-bonded gels under acidic conditions. The addition of NIBA increased the hydrogel modulus and decreased the yield point, suggesting that small NIBA chains strengthened the SA chain network but reduced the flexibility of the SA chains.

The synthesis of double-network hydrogels has been extensively studied, but more in-depth characterization is required. While the rheological behavior of single-network hydrogels is well understood, the understanding of double-network hydrogels is limited. Further research is needed to understand their rheological properties comprehensively. This requires significant effort to achieve a higher level of comprehension in this field.

This paper examines PAAM/SA double-network hydrogels and systematically explores the influence of pH variation, Ca^2+^ ion content, and SA-concentration on both the linear and nonlinear rheological properties and the mechanical characteristics. A thorough state-of-the-art analysis and discussion of the structures would offer a more profound understanding of the material property–rheology relationships.

## 2. Results and Discussion

After the preparation of the PAAM/SA DN hydrogels, they were then characterized by scanning electron microscopy (SEM). The cross-sectional morphology of the pure AM and SA hydrogels is shown in Figure 1a,a’,b,b’, while the PAAM/SA composite hydrogel (samples S1–30) is shown in Figure 1c,c’. The pure PAAM hydrogel has small pores (ca., 10 μm) and a disordered morphology, while pure SA hydrogel appears as a layered structure with layer sizes between 50 μm and 100 μm. In the PAAM/SA hydrogel, SA forms a larger network structure with larger pores than in the pure PAAM network, resulting in more uniform pores and an orderly network structure.

Figure 2b shows the FTIR spectra of AM, SA, and DN PAAM/SA hydrogel samples. For reference, Figure 2c,d illustrates the molecular structure of AM and SA crosslinked by a crosslinking agent, while Figure 2a shows the structure of the double network hydrogel. The FTIR spectrum of PAAM features characteristic absorption peaks at 1605 cm^−1^ and 1608 cm^−1^, denoting an abundance of C=O and N–H moieties of PAAM hydrogels. Meanwhile, for the SA hydrogel, obvious characteristic absorption peaks at 3434 cm^−1^ and 1030 cm^−1^ are observed, indicating that the SA hydrogel contains a large amount of O–H and C–O groups. The characteristic absorption peaks of SA are observed in the PAAM/SA composite hydrogel. The characteristic absorption peaks near 3400 cm^−1^ and 1750 cm^−1^ were significantly enhanced, indicating that the PAAM/SA composite hydrogel system contains many O–H, N–H and C=O groups. Thus, the established system hydrogel comprises a double network structure of SA and PAAM.

### 2.1. Rheological Characterization

Figure 3 displays steady-state shear viscosity analysis of SA and PAAM/SA mixed aqueous solutions at different pHs with the same concentration as the gels but without added crosslinker or Ca^2+^-ions. The concentration of SA used is consistent with the DN hydrogel. The macromolecule SA is composed of M (MMM), G (GGG), or alternating M and G structures, with high viscosity SA polymer chains being unstable and degradable [57]. Figure 3 reveals shear-thinning behavior (lowered viscosity at increased shear rate) stemming from the dissociation of the SA-polymer chains at increased shear stress [85].

The data demonstrate that the sample exhibits gel-like behavior with a yield point at low pH, whereas at high pH, only a slight non-Newtonian behavior is observed for low SA-concentrations (S1). For high SA-concentrations (S2), the shear thinning is much more pronounced, which is easily explainable by the fact that AM is not polymerized and thus behaves more or less like a Newtonian liquid, while SA is a viscoelastic polymer in solution, which leads to shear thinning. While no Ca^2+^-ions were in the system which would have led to supramolecular crosslinking, the moieties of SA can have a whole host of other pH-dependent interactions, especially at low pH [53,57]. Owing to the pH dependence of SA, the pH also has a much more significant influence for S2 than for S1. Obviously, increasing the SA-concentration strongly increases the overall viscosity level, suggesting that the crossover concentration for SA has been exceeded. 

As all the following characterizations were carried out at neutral pH, the data clearly show that no gel-like behavior is found for the precursor solutions. 

Figure 4 shows the results of the creep and creep-recovery test for samples S1 (1.67 wt% SA) and S2 (2.67 wt% SA) [86]. All samples underwent instantaneous deformation (τ = 50 Pa), followed by a much smaller viscoelastic deformation for the remainder of the 300 s creep step, which, as expected, does not converge to a power law slope of 1, which would indicate a Newtonian creep behavior. This shows that the sample is crosslinked, i.e., has an infinite zero shear-rate viscosity [87,88,89]. The recovery step shows that almost the complete deformation is recovered, again indicating that only a very small tendency of permanent (i.e., viscous) deformation is to be found, which is what we would expect from a covalently crosslinked gel. The higher the CaCl_2_-content of the gels, the lower is the residual deformation. This is easily explained by SA being supramolecularly crosslinked by the Ca^2+^-ions. The more Ca^2+^-ions, the higher the probability that the bonding sites on the SA-chain are bonding to a Ca^2+^-ion. However, one should remember that only a bond between one Ca^2+^-ion and 2 SA-bonding sites will lead to physical crosslinking between two chains (both intermolecular intramolecular bonds, while only the intermolecular bonds will contribute to the network) [90,91,92,93]. As the bond between a Ca^2+^-ion and an SA-bonding site is easier to obtain than between Ca^2+^-ion and two SA-bonding sites, too many Ca^2+^-ions will weaken the gel once a significant excess of Ca^2+^-ions is in the system [94,95]. As compliance J and shear modulus G are approximately the inverse of each other, higher strain or compliance J corresponds to a lower modulus [96].

In general, the higher the Ca^2+^-content, the lower is J, i.e., the stiffer the gels, and the lower the residual strain becomes. The differences between the S1 and S2 series appear to be not so large, as the PAAM-network is a major contributor to network strength. At a CaCl_2_ concentration of 0 mg/mL, the final strain values for the creep and recovery phase were significantly higher in S1-0 and S2-0, and the residual strain values were significantly lower than in the other samples. This is because when there is no calcium ion in the sample, SA cannot form a second network, and therefore, only one PAAM network exists in the gel. Thus, the SA is still primarily a viscoelastic polymer solution, which could lead to shear bands and other weak points in the gel, which can be mended by adding Ca^2+^-ions.

Figure 5 and Appendix A (in Appendix A) show the frequency sweeps (FS) for gel sample series S1 and S2, respectively. The storage moduli G′ for all samples except S1-0 and S2-0 is of constant and low slope, while S1-0 and S2-0 have a higher non-constant slope. The loss modulus for those samples is significantly lower and goes through a minimum of around ω = 1 rad/s, which corresponds to loss factor tan δ values below 0.04. Considering that this means δ below 2°, the gels behave close to an ideal rubber [97,98]. Because the rheometer used is a single measurement head rheometer, it is likely that δ is even lower, as for highly elastic soft samples, such rheometers tend to measure too high phase angles.

S1-0 and S2-0 show clearly different dynamic-mechanical data with significantly higher damping and lower overall modulus (especially S2-0). This can be explained by the fact that the PAAM-network is more or less unaffected by the content of Ca^2+^, while the SA-network does not exist, which leads to lots of dangling chains, thus increasing δ and lowering the overall network density, i.e., visible due to a lower G′.

#### Nonlinear Rheological Behavior of (PAAM/SA) Hydrogels 

Figure 6 shows the strain sweeps (γ_0_: 0.01 → 10 → 0.01 → 30 → 0.01 → 100 → 0.01 → 300 → 0.01 → 1000 → 0.01 (%)), whose G′ (Figure 6a) is very similar except for the last leg (γ_0_ = 1000 → 0.01 (%)), which indicates that at γ_0_ > 300% the sample is irreversibly damaged. Figure 6b shows that with each cycle, G″ increases by several % (while G′ is approximately constant). A peak around γ_0_ ≈ 100% appears clearly after the sample is strained with γ_0_ = 300%, which is much larger after γ_0_ = 1000%, and is assigned to damages in the network structure.

Figure 7a,d shows the influence of Ca^2+^-ions on the strain sweeps (γ_0_ = 0.1 → 1000%) for the S1 and S2-series, respectively. The linear viscoelastic regime (G′ and G″ independent of γ_0_) decreases significantly upon the addition of Ca^2+^-ions, while the overall linear viscoelastic values of G′ and G″ as well as the low damping are not affected very much by the presence of Ca^2+^-ions (except S2-0). 

Sim et al. [42] classified the large-amplitude oscillatory shear (LAOS) behavior of strain sweeps into four categories, according to which all samples here behave according to Type III (weak strain overshoot, G′ decreases while G′′ increases and then decreases). With an increasing γ_0_, the sample begins to enter the nonlinear region; thus, the network structure of the gel samples begins to be distorted, resulting in the appearance of weak strain overshoot. The processes occurring here rip open the bonds between Ca^2+^ and the corresponding SA-moieties, which gradually turn the SA-gel network into a polymer solution with dangling ends. Furthermore, the PAAM-network is strained, creating similar effects, albeit without breaking ion–dipole bonds. As γ_0_ increases, the gel samples show a crossover point (G′ = G″). At higher deformations, G′ and G″ decrease sharply, typically with power law slopes of –2 and –1, respectively [99].

In order to better quantify these differences, four characteristic quantities are defined. The linear viscoelastic G′, the modulus at the G′-G″ crossover, the nonlinearity limit, defined by a 5% deviation from the viscoelastic G′, and the deformation at the G′-G″ crossover. Figure 7g,h demonstrates that the influence of ion content on the viscoelastic G′ and modulus at the G′-G″ crossover is relatively small, and the characteristic deformations show a very significant dependence. The γ_0_ corresponding to the nonlinear limit decreased from 5.2% and 13.7% for S1-0 and S2-0, respectively, to 0.1% and 2.37% for S1-5 and S2-5, respectively, followed by a leveling off at concentrations above, ca., 30 mg/mL at γ_0_ ≈ 1% [56,57]. Furthermore, the G″ and deformation $\gamma_{0}$ at the G″-peak were assessed, which, however, yielded almost identical values to the G′-G″-crossover, as those coincide closely. It should be mentioned S1 has significantly lower nonlinearity limits than S2 at low Ca^2+^-concentrations, which we attribute to the lower SA-concentration, leading to a sparse network, which is not fully bis-complexed and thus easy to destroy. The crossover deformation shows a rather similar relationship but without the deep minimum found for the nonlinearity limit.

Figure 7i shows the relation between intrinsic nonlinearity Q_0_ and the calcium ion concentration, which exhibits a jump between 0 and 5 mg/mL Ca^2+^ by 2–3 orders of magnitude, followed by a broad minimum at intermediate concentrations. Knowing that SA has a certain number of bonding moieties and that each of those moieties needs to be complexed with another moiety via a Ca^2+^, it becomes clear that one Ca^2+^ should be in the sample for every 2 SA-moieties. From previous discussions and reports, it is known that more dangling ends in the sample lead to a stronger nonlinearity [83]. Therefore, the closer the concentration of Ca^2+^ is to the optimum concentration of 0.5 equivalent (one Ca^2+^ for every two SA-moieties), the lower *Q*_0_ should be. While the concentration of the SA-moieties is not known precisely, it is clear that their concentration for the S2-series is 60% higher. Taking the minimum position of the S1 and the S2-series, it is clear that the S2-series has a minimum at higher concentration than the S1 series and quantitatively, the factor 1.6 is also justifiable.

The I_3_/I_1_ curves of all gel samples have similar shapes, as shown in Figure 7b,e. In the small strain amplitude region, the waveform of the stress response is consistent with the input strain waveform, thus I_3_/I_1_ → 0 [62]. The I_3_/I_1_ value of each sample increases with an increase in $\gamma_{0}$ until, ca., 10–30% for the samples containing Ca^2+^-ions, followed by a slight decrease and finally, leveling off to the plateau value of S1-0 and S2-0 around I_3_/I_1_ = 0.1. When looking at this peak in detail, it becomes apparent that the peak plateau is virtually reached at the crossover strain of the samples with Ca^2+^-ions, followed by, ca., one order of magnitude before decreasing to the level of the Ca^2+^-free sample (S1-0 and S2-0). This trend is better observed for the S1-series than for the S2-series as the former is “more fragile,” as discussed before.

To the best of our knowledge, such a peak in I_3_/I_1_ has not been reported before, as no research article on double-network hydrogels with supramolecular bonds and LAOS is known to the authors. Apparently, the supramolecular bonds between Ca^2+^ and the corresponding moieties of SA lead to a higher nonlinearity (I_3_/I_1_) than the covalent PAAM-network. Thus, as long as the SA-network is halfway intact, the nonlinearity is higher than when it has broken down due to excessive shear. This could be explained by the supramolecular bonds of SA breaking and reforming twice within one cycle, which will lead to a stronger deviation from the sinusoidal curve shape. This fundamental difference in the response caused by Ca^2+^-ions is also clearly seen in the viscous Lissajous plots of S1 (Figure 8), other Lissajous plots can be found in Appendix A, where the curve shapes of $\gamma_{0}$ = 0.1%, 300%, and 1000% are rather similar, while for the other deformations shown, S1-0 is clearly different from the other samples. 

Figure 7c,f shows the *Q*-parameters, which, just like I_3_/I_1_, are almost indistinguishable for γ_0_ > 300%, while for smaller deformations, the samples without Ca^2+^-ions show a significantly higher nonlinearity Q than S1-0 and S2-0. From these data, the intrinsic nonlinearity parameter Q_0_ was determined.

The Chebyshev coefficients, strain hardening rate (S), and shear thickening rate (T), as defined in the experimental part, can be used to analyze the nonlinear rheological behavior of gel samples in LAOS experiments [100]. In LAOS testing, the third Chebyshev coefficients (e_3_ and v_3_) are mainly used to characterize the nonlinear response, and their physical interpretation is related to S and T: when e_3_/e_1_ > 0, the sample exhibits strain hardening properties similar to S > 0; when e_3_/e_1_ < 0, the sample exhibits strain softening properties similar to S < 0; when e_3_/e_1_ = 0, the gel sample is in the LVR region. When v_3_/v_1_ > 0, the sample exhibits shear thickening properties similar to T > 0; when v_3_/v_1_ < 0, the sample exhibits shear thinning properties similar to T < 0; when v_3_/v_1_ = 0, the gel sample is in the LVR region.

As shown in Figure 9a,c,e and Appendix A, as γ_0_ increases, samples S1 and S2 undergo a transition from the LVR region to the nonlinear viscoelastic region. During this transition, samples S1-(0, 20–80)/S2-(0, 20–80) exhibit strain softening before strain hardening, while sample S1-(10–15)/S2-(10–15) directly undergoes strain hardening without any strain softening. This indicates that the samples S1-(10–15)/S2-(10–15) have a faster feedback response to strain and stress and can immediately respond when subjected to γ_0_ beyond the maximum tolerance of the LVR region, which is also related to its higher strength and G′. The onset of strain hardening indicates that the gel samples have entered the nonlinear viscoelastic region. The reason for the strain hardening of gel samples is that when the gel sample is subjected to γ_0_, the network structure of the sample can resist deformation. However, when the γ_0_ exceeds a certain threshold, the network structure of the gel sample is destroyed, and the ability to resist deformation decreases significantly. This can also explain the phenomenon of S peaking, followed by decreasing at large deformations.

Furthermore, it should be pointed out that compared to I_3_/I_1_, the sensitivity to changes in structure is higher, as the deformation at which S, T, e_3_/e_1_, or v_3_/v_1_ deviate significantly from the linear values varies systematically stronger—e.g., when looking at the strain γ_0_, at which S = 0.5, a relation very similar to Figure 7i would be obtained. 

The presence of a peak in I_3_/I_1_, S, T, e_3_/e_1_, or v_3_/v_1_ can be interpreted as the fingerprint of the SA-network breakdown and reformation, which is possible due to its supramolecular nature. In the future, observations on other systems should be able to highlight whether this is a reliable fingerprint of covalent-supramolecular double-network gels.

### 2.2. Mechanical Properties 

Figure 10 shows the mechanical properties characterization of all hydrogel samples. Figure 10a,c display nominal stress–strain curves. Hydrogel samples without calcium ions are relatively weak and brittle. However, when a strong SA network is formed by adding an appropriate amount of calcium ions, the fracture strength and strain of the hydrogel samples increase sharply, reaching a maximum of 0.33 MPa and 700%, respectively. 

Figure 10b,d shows the relationship between the tensile strength and toughness of the samples and the concentration of calcium ions. The toughness of the gel samples is defined as the area under the stress–strain curve of the sample up to the point where it fractures. As shown in Figure 9b,d, the tensile strength and toughness of sample series S1 and S2 initially increase and then decrease as the concentration of calcium ions increases. This is because calcium ions coordinate with the M and G blocks on the alginate during gelation, acting as connections between adjacent blocks (i.e., the egg-box model) [101]. The semi-rigid chains of G-block-containing alginate strongly interact electrostatically with calcium ions. Thus, the SA network structure adds a higher strength than AM chains possess alone. When the sample is stretched, the SA-Ca^2+^-SA bonds are the first to break, potentially followed by a new bond being established (sticky reptation) [102,103,104]. Suppose there are too many or too few Ca^2+^-ions relative to the number of carboxylic acid moieties on the SA, either un-complexed Ca^2+^-ions or mono-complexed Ca-SA-bonds exist, and both are non-bonding. Stretching the sample and, thus, breaking the SA-Ca^2+^-SA-bonds leads to a reformation of these bonds. However, as non-bonding mono-complexes are easier to form compared to bonding bis-complexes and an equilibrium between mono-and bis-complexes exists, it is logical that potentially un-complexed Ca^2+^-ions tend to form mono-complexes more likely than mono-complexes tend to form bis-complexes if an excess of Ca^2+^-ions is present in the sample. 

The optimum concentration of Ca^2+^-ions for the S1-series is 10 mg/mL, while for S2 there is a broader maximum with 20 and 30 mg/mL being approximately equal. The factor 1.6 mentioned above in SA-concentration becomes apparent here again. Overall, the increase of tensile strength and toughness of the best samples of S1 vs. S2-series (ca. 50%) is significantly higher than the increase of the total amount of polymer in the sample (S1: 16.67 wt% AM + 1.667 wt% SA vs. S2: 16.67 wt% AM + 2.667 wt% SA) by only 6%. This can be explained by the nature of double-network hydrogels, where one network stabilizes the other network. Clearly, a very low SA-concentration will lead to an improvement of the mechanical properties, but a higher concentration of S2 will lead to a stronger second network. 

## 3. Experimental Part

### 3.1. Materials

*Preparation of PAAM/SA composite hydrogel*: The composite hydrogel was prepared by a two-step method. Different ratios of AM (acrylamide, 99%, Aladdin, Shanghai, China) and SA (sodium alginate, AR, Aladdin, Shanghai, China) monomers were dissolved in deionized water, followed by the addition of N,N′-methylene bisacrylamide (MBAA, AR, Macklin, Shanghai, China) as crosslinker and tetramethyl-ethylenediamine (TEMED, 99% Macklin, Shanghai, China) as accelerator, mixed into a homogeneous precursor solution by ultrasonic and magnetic stirring. The precursor solution contained 0.1 g/L of MBAA and 50 μL of 99% TEMED; the concentrations of the different monomers added are shown in Table 1. Then, 40 μL of 0.05 wt% potassium persulfate (KPS, 99.9%, McLean, Shanghai, China) was added as an initiator to the homogeneous precursor solution and poured into the mold (40.0 × 40.0 × 2.0 mm^3^). The gel was cured at 60 °C for one hour, during which the AM monomers formed the first covalent 3-dimensional network. Aqueous calcium chloride solution (CaCl_2_, 99.9%, Aladdin, Shanghai, China) was used to displace the aqueous solution in the hydrogel over 3 days. This solution exchange led to the formation of the second network SA, and the formation of PAAM/SA composite hydrogels was accomplished. For comparison, a PAAM gel was prepared in the same way as the composite hydrogel (except that no SA monomer was added). A pure SA gel was prepared by adding calcium chloride solution to the SA aqueous solution, followed by mixing and stabilizing for 24 h. All hydrogel samples discussed in this paper are listed in Table 1.

### 3.2. Characterization

#### 3.2.1. SEM and FTIR

For SEM (scanning electron microscopy), the hydrogel was first immersed in liquid nitrogen for 10 min, followed by freeze-drying for two days. Then, the sample was broken to create a clean surface. Subsequently, the sample was mounted on an SEM stage with conductive carbon tape and a thin layer of gold was sputtered at 20 mA under vacuum for 50 s. The cross-sectional morphology of the hydrogel was observed using SEM. The prepared hydrogel was characterized using SEM (SU-70, Hitachi, Tokyo, Japan). The Fourier transform infrared (FTIR) spectra of the DN hydrogel were collected using a Nicolet 6700 spectrometer (Thermo Fisher Scientific, Waltham, MA, USA) with KBr pellets. The samples were prepared by grinding and dispersing the freeze-dried sample in KBr, pressing them into pellets, and performing infrared characterization. The spectral range was 400–4000 cm^−1^ with a resolution of 4 cm^−1^.

#### 3.2.2. Rheology

Rheological characterization of two types of double-network hydrogels (S1 and S2) was performed using an Anton Paar MCR 302 rheometer (Graz, Austria). A parallel plate geometry with a 25 mm diameter plate (PP 25) was used at a test temperature of 25 °C and a 2 mm gap between the plates for hydrogel samples. Before testing, the samples were allowed to equilibrate for five minutes to ensure complete relaxation. For precursor solutions, cone-and-plate geometry with a 50 mm diameter plate and a 1° angle (CP 50/1) was used with a 0.102 mm gap between the plates to prevent non-uniform deformation and flow of the sample. 

The steady-state viscosity of the samples was measured at shear rates $\dot{\gamma }$ of 0.01–1000 1/s after pre-shearing to eliminate the effects of air bubbles and non-uniform distribution [105]. The viscosity functions η($\dot{\gamma }$) were used for the liquid samples to determine the rheological behavior at different pH values. Creep recovery experiments were conducted to determine the changes in strain and creep compliance over time [68]. A constant stress τ of 50 Pa (in the linear viscoelastic region (LVR)) was applied for 300 s, followed by a creep recovery experiment for 600 s (stress τ = 0 Pa). While this setup does not completely ensure the stationarity of the deformation, it is a good indicator of the gels being viscoelastic liquids or solids. Frequency sweep tests were conducted at ω = 100–0.1 rad/s with a (strain $\gamma_{0}$ of 1% in the linear viscoelastic regime) to study the frequency dependence of the composite hydrogel’s linear viscoelastic region and monitor the storage modulus (G′), loss modulus (G″), and loss factor (tan δ) as a function of frequency ω [106].

To characterize the linear and nonlinear viscoelastic regions of the gel samples, a dynamic strain sweep (DSS) was performed by increasing and decreasing the strain values logarithmically at a constant angular frequency ω = 1 rad/s. DSS was performed with five continuous cycles of increasing and decreasing strain amplitude $\gamma_{0}$ (Table 2). 

While in the linear viscoelastic regime G′, G″, δ, … are clearly defined, the occurrence of overtones in the nonlinear regime requires a more detailed analysis. For this purpose, the data were analyzed with MITLAOS [40,60], yielding the intensities of each (odd) overtone I_n_ as well as the corresponding phase angles δ_n_, from which the Chebichev coefficients e_n_ and v_n_ as well as the higher order components of the moduli G′_n_ and G″_n_ were calculated.

In LAOS, the stress response is composed of odd higher harmonics, and the I_n_/I_1_ ratio is the most commonly used indicator for quantifying and predicting nonlinear behavior. The signal-to-noise ratio (SNR), which is the ratio of the amplitude of the first harmonic peak to the standard deviation of the noise, is typically between 10^3^ and 10^5^. In addition to the higher harmonic ratio, the nonlinear parameter Q can also be used to characterize the degree of nonlinearity of the gel sample [57]: (1)$$Q=\frac{I_{3}/I_{1}}{{\gamma_{0}}^{2}}$$

This is used to determine Q_0_, the Q-parameter for $\gamma_{0}$ → 0 [107,108]. Ewoldt et al. [40] introduced the minimum-strain modulus G′_M_, the large-strain modulus G′_L_, defined as:(2)$$G_{M}^{\prime }=\sum_{n=\mathrm{odd}}n{G\prime }_{n},G_{L}^{\prime }=\sum_{n=\mathrm{odd}}{{G^{\prime }}_{n}(-1)}^{(n-1)/2}$$

As well as the minimum-rate dynamic viscosity η′_M_, the large-rate dynamic viscosity η′_L_, is defined as:(3)$$\eta_{L}^{\prime }=\frac{1}{\omega }\sum_{n=\mathrm{odd}}{G\prime \prime }_{n},\eta_{M}^{\prime }=\frac{1}{\omega }\sum_{n=\mathrm{odd}}{G\prime \prime }_{n}{(-1)}^{(n-1)/2}$$

From the minimum-strain modulus, the large-strain modulus, the minimum-rate dynamic viscosity, and large-rate dynamic viscosity, the strain hardening (S), and shear-thickening (T) were determined. These properties can also be visually obtained from the Lissajous curves [64]:(4)$$S\left(\omega ,\gamma_{0}\right)=\frac{G_{L}^{\prime }\left(\omega ,\gamma_{0}\right)-G_{M}^{\prime }\left(\omega ,\gamma_{0}\right)}{G_{L}^{\prime }\left(\omega ,\gamma_{0}\right)}$$
(5)$$T\left(\omega ,\gamma_{0}\right)\frac{\eta_{L}^{\prime }\left(\omega ,\gamma_{0}\right)-\eta_{M}^{\prime }\left(\omega ,\gamma_{0}\right)}{\eta_{L}^{\prime }\left(\omega ,\gamma_{0}\right)}$$

When S < 0, the sample exhibits strain-softening behavior, while when S > 0, the sample exhibits strain-hardening behavior. When T < 0, the sample exhibits shear-thinning behavior, while when T > 0, the sample exhibits shear-thickening behavior.

#### 3.2.3. Tensile Testing

Gel samples (25.0 × 20.0 × 2.0 mm^3^) were tested for uniaxial stretching at a constant rate of 50 mm/min using a universal testing machine (CMT 5000, Sansi Testing, Shenzhen, China). The samples were cut into a dumbbell shape. The stress–strain curves were used to display the corresponding toughness [109], with nominal stress σ calculated as the ratio of tensile force σ to the cross-sectional area A of the deformed sample [110]. The mechanical properties of all samples were calculated as the average of at least three specimens. For cyclic tensile curves, samples were loaded at a predetermined strain ratio, and immediately unloaded, and the hysteresis loop area estimated by the loading and unloading curves was used to calculate the sample’s hysteresis energy density [111,112,113].

## 4. Conclusions

The rheological behavior of PAAM/SA composite hydrogels was studied under large strains using SAOS and LAOS rheological methods. Microstructure changes were analyzed through mechanical properties characterization, as well as physical, and morphological analysis. 

Both PAAM/SA gel precursor solutions and AM and SA aqueous solutions were in sol state under alkaline and natural conditions and in gel state under acidic conditions. Higher concentrations of SA in the solution caused more interactions between SA chains. After AM penetrated the SA chain, the SA chain needed more energy dissipation to be completely deconstructed.

From the linear rheological characterization, it was found that the addition of the SA rigid network structure significantly increased the G′ (from ≈ 4 to 13 kPa to ≈ 10 to 20 kPa) and the strength of the gel, since the PAAM network was a tough network. In the nonlinear rheological characterization, all gel samples showed Type III behavior, which means weak strain overshoot (G′ decreases with increasing strain, and G″ increases and then decreases). This weak strain overshoot indicates that the gel sample could resist deformation. The response of the single network PAAM hydrogel to strain was later and smaller in the nonlinear region, having a nonlinearity limit at, ca., 10% deformation, typical for polymer melts and solutions, vs. ca. 0.1–2% for the double network gels. The response of the PAAM/SA composite hydrogel to strain was faster, with the nonlinear transition occurring more quickly and the nonlinear region being larger. From the nonlinear parameters, all gel samples initially exhibited strain hardening in the nonlinear region, followed by shear thickening at intermediate strains, and shear banding as the strain increased, which can be seen from the nonlinear parameters S, T, e_3_/e_1_, and v_3_/v_1_, exhibiting a very unusual peak or double-peak (v_3_/v_1_) at intermediate $\gamma_{0}$. The reason for the shear banding is that the network structure in the gel sample is destroyed sequentially. Around $\gamma_{0}$ = 10–100%, the alginate network becomes destructured, while deformations above $\gamma_{0}$ = 100% are necessary for destructuring the AA network. Finally, the gel sample exhibited the same response as the strain continued to increase.

The high-order harmonic to the fundamental harmonic ratio (I_n_/I_1_), nonlinear parameters (Q, e_3_/e_1_, v_3_/v_1_), and Chebyshev coefficients (S, T) are more sensitive to the interaction of polymers in hydrogels. The nonlinear viscoelasticity under dynamic oscillatory shear, particularly at high strain, helps study the network structure and linear/nonlinear viscoelastic behavior of hydrogel systems, providing guidance for their design and processing.

The tensile strength and toughness of the PAAM/SA composite hydrogel both showed a trend of first increasing and then decreasing with an increase in the concentration of calcium ions. The toughness of S1-0 and S2-0 is very low (below 0.02 kJ/m^3^), while it can reach a maximum of 0.24–0.34 kJ/m^3^ at intermediate concentrations of Ca^2+^ before decreasing at higher concentrations. This indicates that there is a threshold for the concentration of calcium ions that enhances the strength of the gel sample, which is related to the concentration of SA in the sample. The amount of calcium ions required for the physical cross-linking of SA chains in the gel sample is fixed by the concentration of calcium-ion-accepting moieties. When the concentration of calcium ions exceeds this threshold, the calcium ion strengthening of the gel sample by the Ca^2+^ ions will be reduced due to an oversupply of calcium ions, which lead sto a shift of the physicochemical balance to monocomplexes, which do not have linking chains.

This article focuses on the rheological characterization of hydrogels as soft matter, studying chain dynamics and nonlinear rheological effects under large strains, specifically in double-network hydrogels. 

The results of this paper could contribute to further understanding of the mechanics of double-network hydrogels and thus help design strong hydrogels for future soft matter applications.

## Figures and Tables

**Figure 1 molecules-28-04868-f001:**
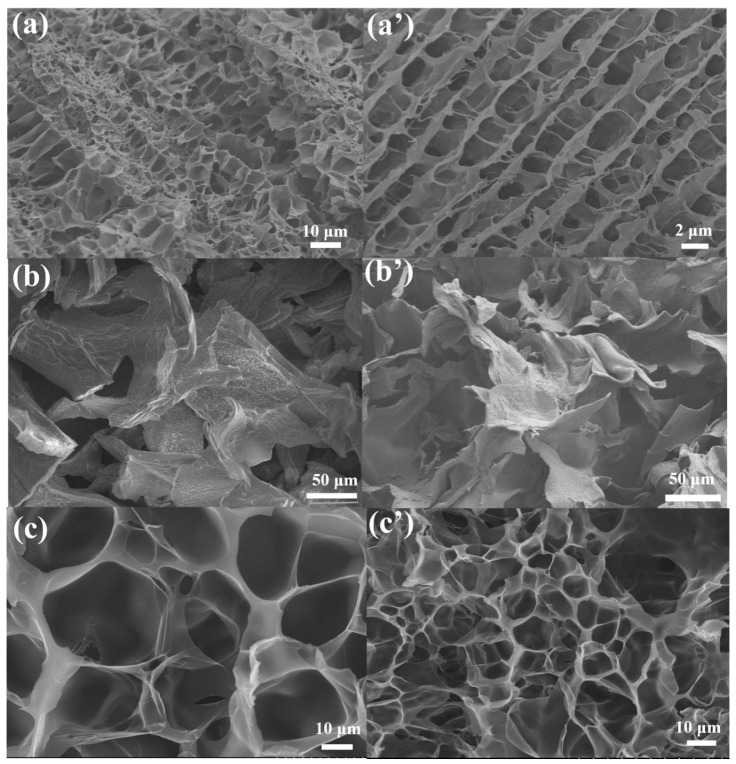
SEM characterization of gel samples: (**a**,**a’**) pure acrylamide hydrogel, (**b**,**b’**) pure SA hydrogel, (**c**,**c’**) PAAM/SA composite hydrogel (samples S1–30).

**Figure 2 molecules-28-04868-f002:**
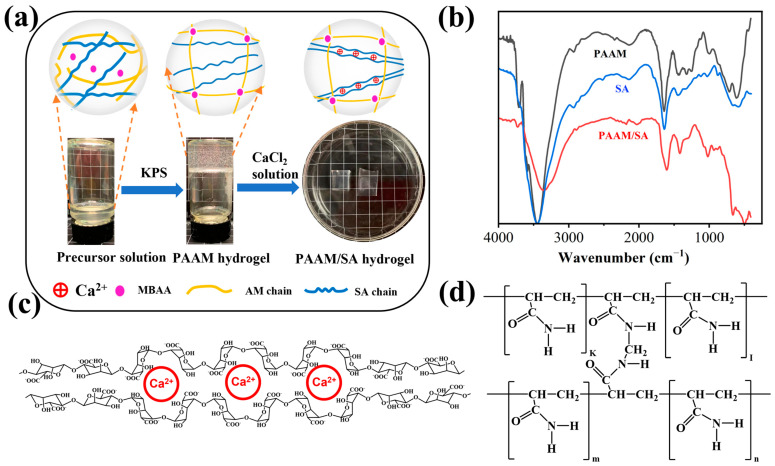
(**a**) Schematic diagram of sample preparation and internal network structure, (**b**) FTIR analysis of three gel samples, (**c**) SA chain segments supramolecularly “cross-linked” by calcium ions, (**d**) interactions between AM monomer and MBAA.

**Figure 3 molecules-28-04868-f003:**
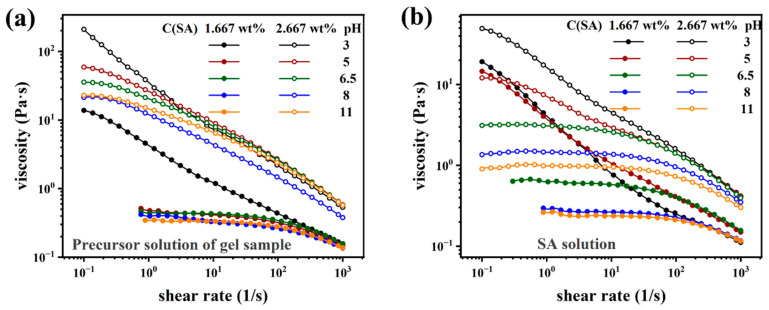
(**a**) Viscosity characterization of sample S1 (1.67 wt% SA) and S2 (2.67 wt% SA) precursor solutions at different pHs. (**b**) Viscosity characterization of SA aqueous solution at different pHs.

**Figure 4 molecules-28-04868-f004:**
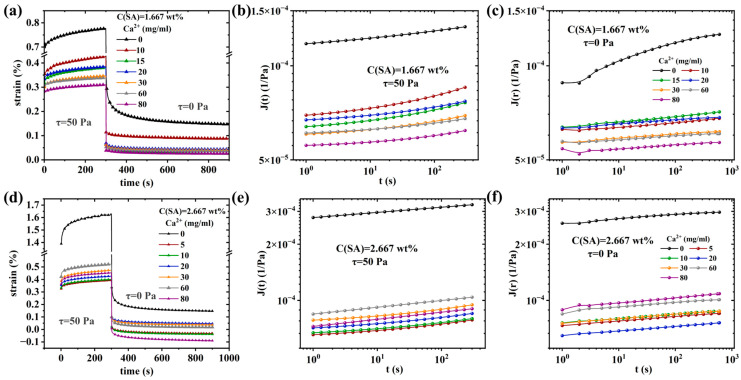
Creep recovery behavior of sample: (**a**) S1 (SA = 1.667 wt%), (**b**) S2 (SA = 2.667 wt%). (**a**,**d**) Strain vs. time, (**b**,**c**,**e**,**f**) log creep and creep recovery compliance vs. log time.

**Figure 5 molecules-28-04868-f005:**
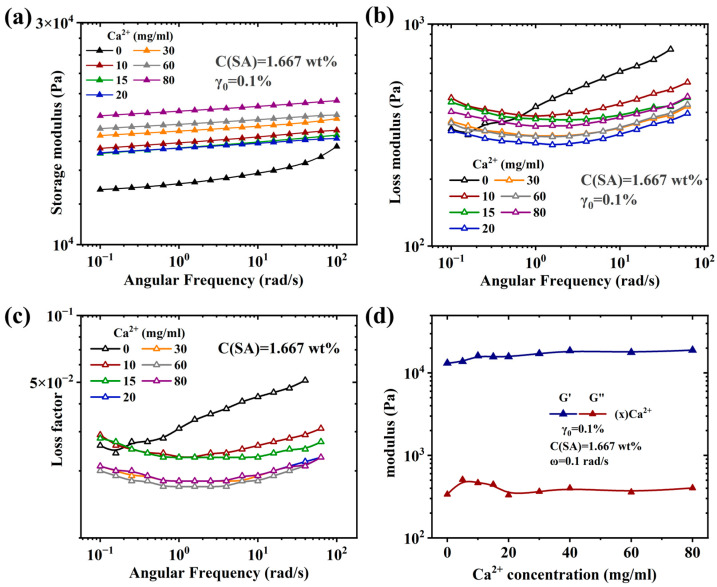
(**a**,**b**) Frequency sweep curve of sample S1 showing the storage (**a**) and the loss modulus (**b**). (**c**) Relationship between loss coefficient and angular frequency, and (**d**) relationship between the modulus of sample S1 and calcium ion concentration.

**Figure 6 molecules-28-04868-f006:**
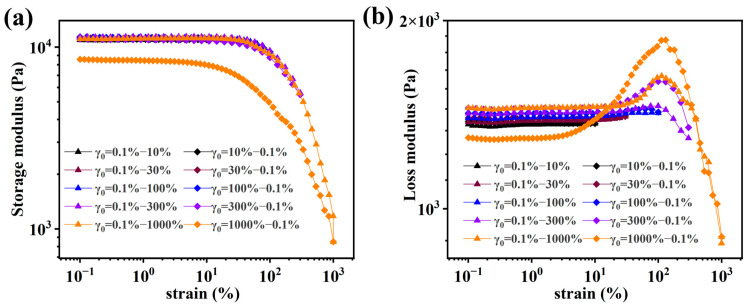
(**a**,**b**) Strain sweeps of S1-0 with increasing and decreasing strain amplitude $\gamma_{0}$.

**Figure 7 molecules-28-04868-f007:**
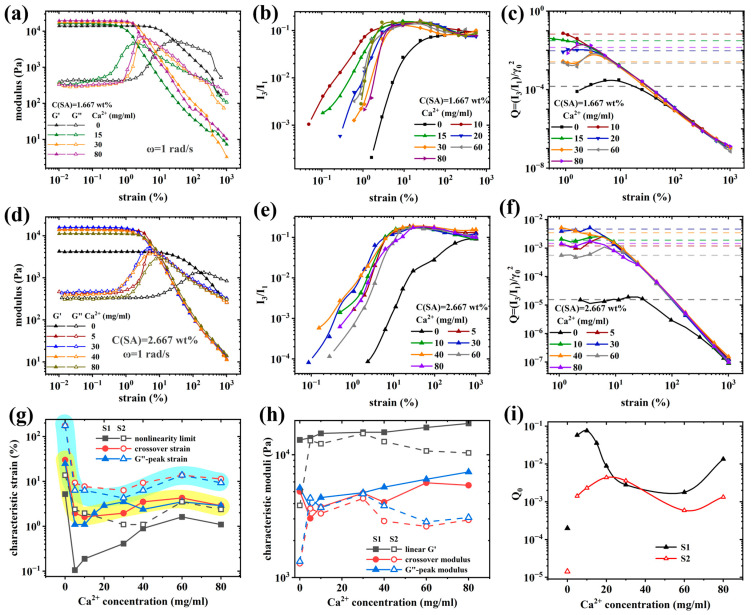
Dynamic strain sweep curves and analysis of nonlinear parameters for gel samples with different calcium ion concentrations. DSS curves: (**a**) sample S1, (**d**) sample S2. Analysis of nonlinear parameters I_3_/I_1_ (**b**,**e**) and Q (**c**,**f**): (**b**,**c**) sample S1, (**e**,**f**) sample S2. (**g**,**h**) Characteristic coefficients determined by nonlinear limits of S1 and S2. (**i**) Relationship between intrinsic nonlinearity Q_0_ and calcium ion concentration.

**Figure 8 molecules-28-04868-f008:**
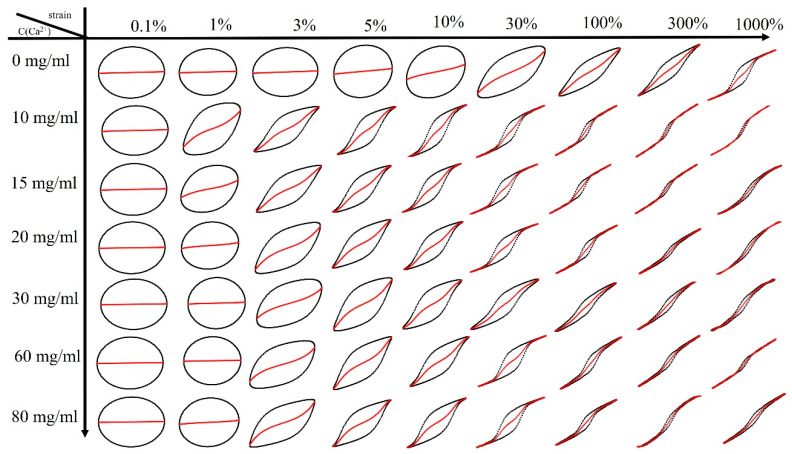
Viscous Lissajous curves of sample S1 at varying calcium ion concentrations with different γ_0_.

**Figure 9 molecules-28-04868-f009:**
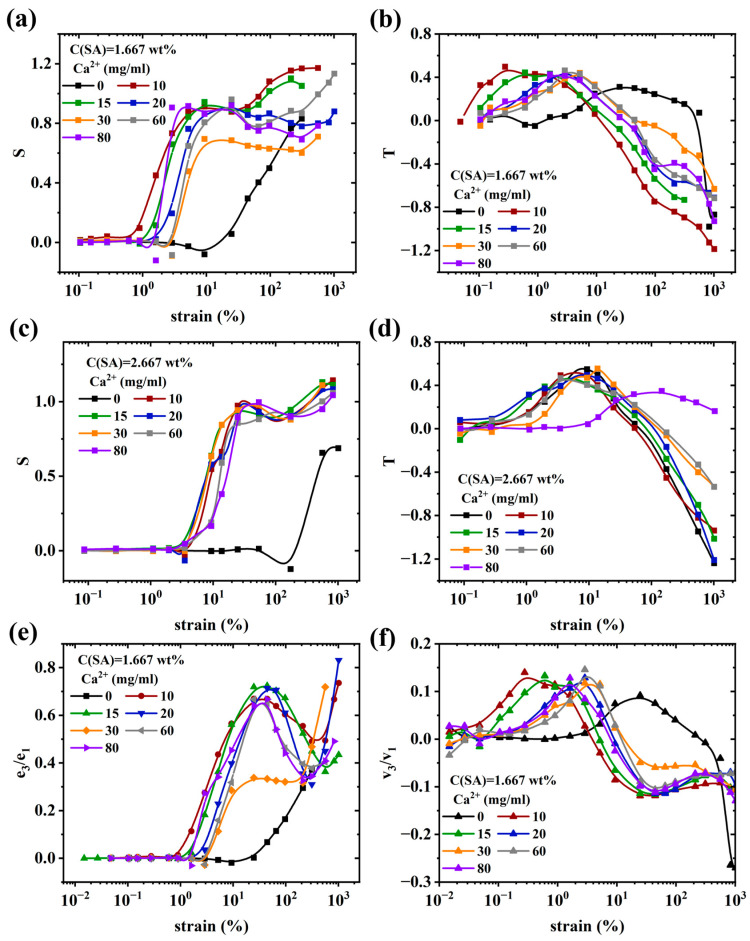
(**a**) Strain hardening ratio S, (**b**) shear thickening ratio T versus strain for sample S1. (**c**) Strain hardening ratio S, (**d**) shear thickening ratio T versus strain for sample S2. (**e**,**f**) Chebyshev coefficient (e_3_/e_1_, and v_3_/v_1_) versus strain for sample S1.

**Figure 10 molecules-28-04868-f010:**
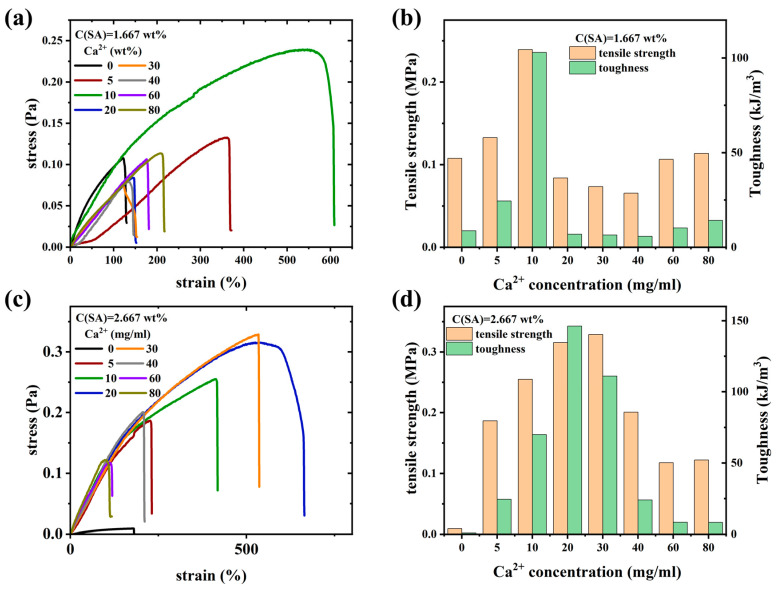
Tensile stress–strain curves of the S1- (**a**) and S2-series (**c**), corresponding tensile strengths and toughnesses (**b**,**d**) at different calcium ion concentrations.

**Table 1 molecules-28-04868-t001:** Names and monomer concentrations of all samples used in this study.

Sample Name	AM (wt%)	SA (wt%)	MBAA (wt%)	TEMED, (wt%)	KPS (wt%)	X (mg/mL) = C(Ca^2+^)
S1-X	16.67	1.667	0.01	0.167	0.067	0, 5, 10, 20, 30, 40, 60, 80
S2-X	16.67	2.667	0.01	0.167	0.067	0, 5, 10, 20, 30, 40, 60, 80

**Table 2 molecules-28-04868-t002:** Dynamic strain sweep parameters.

	Interval 1	Interval 2	Interval 3	Interval 4	Interval 5
$\gamma_{0}$ (%)	0.01 → 10 10 → 0.01	0.01 → 30 30 → 0.01	0.01 → 100 100 → 0.01	0.01 → 300 300→ 0.01	0.01 → 1000 1000 → 0.01
ω (rad/s)	1	1	1	1	1

## Data Availability

The raw data are available from the authors upon request.

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
