# Peer review of "Understanding Gel-Powers: Exploring Rheological Marvels of Acrylamide/Sodium Alginate Double-Network Hydrogels"

_molecules, 2023, doi:10.3390/molecules28124868_

Round 1

Reviewer 1 Report

Dear Authors,

I studied your manuscript entitled "Unlocking Gel-powers: Exploring Rheological Marvels of Acrylamide/Sodium Alginate Double-Network Hydrogels". This paper comprises interesting results that certainly deserve publication. I recommend a major revision before further consideration for publication in the Molecules.

1) The quality of the abstract and conclusion should be improved by inserting key results of the work. It would be helpful if these sections contained some more quantitative data.

2) How do LAOS and SAOS testing differ in their ability to characterize the rheological properties of hydrogels, and what are some of the limitations of these techniques?

3) The introduction provides a lot of technical information on IPN hydrogels, but it's not immediately clear why IPN hydrogels are relevant to the research question. Can you clarify the connection between IPN hydrogels and their research question?

4) What is the significance of the peak in I3/I1 in the double-network hydrogels with supramolecular bonds and LAOS? How does this contribute to the field?

5) Does calcium ion-containing hydrogel have different mechanical properties from those without calcium ions? Why is this difference significant? What is the effect of calcium ion concentration on the mechanical properties of hydrogels? Which concentration is optimal for each series of samples?

6) Using LAOS measurements, how do you ensure that they are reliable and accurate? Could the results be affected by any potential sources of error?

7) What is the significance of the egg-box model for the coordination of calcium ions with the polymer chains? How does this model explain the observed increase in mechanical strength and toughness of the hydrogels with increasing calcium ion concentration?

8) In practical applications, how do you deal with potential limitations or challenges, ethical concerns, and safety concerns, such as environmental conditions and potential toxicity concerns?

9) In your opinion, what is the relationship between hydrogel mechanical properties and LAOS measurements? In the context of rheology and materials science, how do you interpret the LAOS measurements? How do these measurements provide new insights?

10) English language needs some polishing since some terms are vague. The paper's title is also recommended to be revised.

English language needs some polishing since some terms are vague. 

Reviewer 2 Report

The article is made at a good scientific and technical level, and its practical significance is beyond doubt. In order to improve the readability and clarity of the manuscript, some major concerns need to be addressed before the paper is to be processed further:
1. The abstract requires some brief quantitative results. The abstract is a mini version of the manuscript that proceeds. So, include the introduction, methodology, results and concluding remarks in a precise but effective manner.
2. The aim of this investigation must be mentioned clearly at the end of the introduction, specifying this work's objective and novelty.
3. The motivation for the study and the research gap needs to be clarified.
4. Discussion needs a more scientific explanation for the obtained results. Authors should attribute the results achieved to a clear scientific reason.
5. More relevant references should be incorporated in the introduction section. Cite the following paper:

Dogra, V., Kishore, C., Mishra, A., Gaur, A. and Verma, A., 2023. Sol-Gel preparation and wetting behaviour analysis of hydrophobic Zirconium based nano-coating: Implications for solar panel coating. Chemical Engineering Journal Advances, 15, p.100507.

1. The English language used in the paper is to be revised and improved before the subsequent manuscript submission. Please, read the text carefully before the subsequent submission of this paper.

Reviewer 3 Report

The topic of the paper matches the Aims of Molecules Journal. The paper mainly focuses on rheological characterization of DN Hydrogels made of covalent crosslinked polyacrylamide and supramolecular crosslinked sodium alginate. 

I think the work is well presented and is scientifically sound. 

I would only recommend to double check the syntax of sentences.

Minor English improvements, to improve the flow of the manuscript.

Round 2

Reviewer 1 Report

Dear Authors,

Thank you for considering my comments. I have recommended the publication of your article as is.

Reviewer 2 Report

Accept as is.